# Hierarchical interpretations for neural network predictions

**Chandan Singh**[*]
Department of EECS
UC Berkeley
c_singh@berkeley.edu

**W. James Murdoch**[*]
Department of Statistics
UC Berkeley
jmurdoch@berkeley.edu

**Bin Yu**
Department of Statistics, EECS
UC Berkeley
binyu@berkeley.edu

## Abstract

Deep neural networks (DNNs) have achieved impressive predictive performance due to their ability to learn complex, non-linear relationships between variables. However, the inability to effectively visualize these relationships has led to DNNs being characterized as black boxes and consequently limited their applications. To ameliorate this problem, we introduce the use of hierarchical interpretations to explain DNN predictions through our proposed method: agglomerative contextual decomposition (ACD). Given a prediction from a trained DNN, ACD produces a hierarchical clustering of the input features, along with the contribution of each cluster to the final prediction. This hierarchy is optimized to identify clusters of features that the DNN learned are predictive. We introduce ACD using examples from Stanford Sentiment Treebank and ImageNet, in order to diagnose incorrect predictions, identify dataset bias, and extract polarizing phrases of varying lengths. Through human experiments, we demonstrate that ACD enables users both to identify the more accurate of two DNNs and to better trust a DNN's outputs. We also find that ACD's hierarchy is largely robust to adversarial perturbations, implying that it captures fundamental aspects of the input and ignores spurious noise.

## 1 Introduction

Deep neural networks (DNNs) have recently demonstrated impressive predictive performance due to their ability to learn complex, non-linear, relationships between variables. However, the inability to effectively visualize these relationships has led DNNs to be characterized as black boxes. Consequently, their use has been limited in fields such as medicine (e.g. medical image classification (Litjens et al., 2017)), policy-making (e.g. classification aiding public policy makers (Brennan & Oliver, 2013)), and science (e.g. interpreting the contribution of a stimulus to a biological measurement (Angermueller et al., 2016)). Moreover, the use of black-box models like DNNs in industrial settings has come under increasing scrutiny as they struggle with issues such as fairness (Dwork et al., 2012) and regulatory pressure (Goodman & Flaxman, 2016).

To ameliorate these problems, we introduce the use of hierarchical interpretations to explain DNN predictions. Our proposed method, agglomerative contextual decomposition (ACD)[1], is a general technique that can be applied to a wide range of DNN architectures and data types. Given a prediction from a trained DNN, ACD produces a hierarchical clustering of the input features, along with the contribution of each cluster to the final prediction. This hierarchy is optimized to identify clusters of features that the DNN learned are predictive (see Fig 1).

---

[*]Equal contribution, order determined by coin flip

[1]Code and scripts for running ACD and experiments available at https://github.com/csinva/acd

The development of ACD consists of two novel contributions. First, importance scores for groups of features are obtained by generalizing contextual decomposition (CD), a previous method for obtaining importance scores for LSTMs (Murdoch et al., 2018). This work extends CD to arbitrary DNN architectures, including convolutional neural networks (CNNs). Second, most importantly, we introduce the idea of hierarchical saliency, where a group-level importance measure, in this case CD, is used as a joining metric in an agglomerative clustering procedure. While we focus on DNNs and use CD as our importance measure, this concept is general, and could be readily applied to any model with a suitable measure for computing importances of groups of variables.

We demonstrate the utility of ACD on both long short term memory networks (LSTMs) (Hochreiter & Schmidhuber, 1997) trained on the Stanford Sentiment Treebank (SST) (Socher et al., 2013) and CNNs trained on MNIST (LeCun, 1998) and ImageNet (Russakovsky et al., 2015). Through human experiments, we show that ACD produces intuitive visualizations that enable users to better reason about and trust DNNs. In particular, given two DNN models, we show that users can use the output of ACD to select the model with higher predictive accuracy, and that overall they rank ACD as more trustworthy than prior interpretation methods. In addition, we demonstrate that ACD's hierarchy is robust to adversarial perturbations (Szegedy et al., 2013) in CNNs.

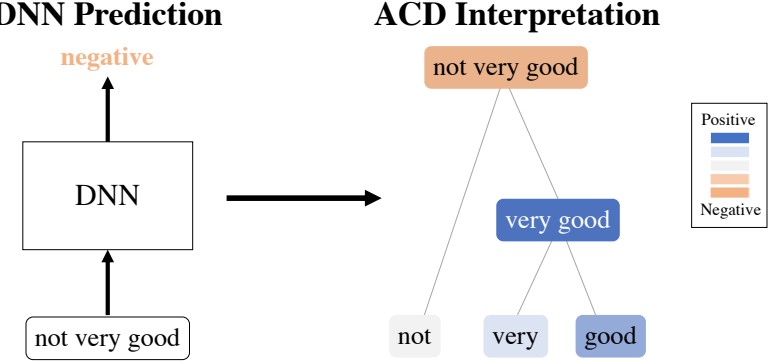

Figure 1: ACD illustrated through the toy example of predicting the phrase "not very good" as negative. Given the network and prediction, ACD constructs a hierarchy of meaningful phrases and provides importance scores for each identified phrase. In this example, ACD identifies that "very" modifies "good" to become the very positive phrase "very good", which is subsequently negated by "not" to produce the negative phrase "not very good". Best viewed in color.

## 2 Background

Interpreting DNNs is a growing field (Murdoch et al., 2019) spanning a range of techniques including feature visualization (Olah et al., 2017; Yosinski et al., 2015), analyzing learned weights (Tsang et al., 2017) and others (Frosst & Hinton, 2017; Andreas et al., 2016; Zhang et al., 2017). Our work focuses on local interpretations, where the task is to interpret individual predictions made by a DNN.

**Local interpretation**   Most prior work has focused on assigning importance to individual features, such as pixels in an image or words in a document. There are several methods that give feature-level importance for different architectures. They can be categorized as gradient-based (Springenberg et al., 2014; Sundararajan et al., 2017; Selvaraju et al., 2016; Baehrens et al., 2010), decomposition-based (Murdoch & Szlam, 2017; Shrikumar et al., 2016; Bach et al., 2015) and others (Dabkowski & Gal, 2017; Fong & Vedaldi, 2017; Ribeiro et al., 2016; Zintgraf et al., 2017), with many similarities among the methods (Ancona et al., 2018; Lundberg & Lee, 2017).

By contrast, there are relatively few methods that can extract the interactions between features that a DNN has learned. In the case of LSTMs, Murdoch et al. (2018) demonstrated the limitations of prior work on interpretation using word-level scores, and introduced contextual decomposition (CD), an algorithm for producing phrase-level importance scores from LSTMs. Another simple baseline is occlusion, where a group of features is set to some reference value, such as zero, and the importance

of the group is defined to be the resulting decrease in the prediction value (Zeiler & Fergus, 2014; Li et al., 2016). Given an importance score for groups of features, no existing work addresses how to search through the many possible groups of variables in order to find a small set to show to users. To address this problem, this work introduces hierarchical interpretations as a principled way to search for and display important groups.

**Hierarchical importance**   Results from psychology and philosophy suggest that people prefer explanations that are simple but informative (Harman, 1965; Read & Marcus-Newhall, 1993) and include the appropriate amount of detail (Keil, 2006). However, there is no existing work that is both powerful enough to capture interactions between features, and simple enough to not require a user to manually search through the large number of available feature groups. To remedy this, we propose a hierarchical clustering procedure to identify and visualize, out of the considerable number of feature groups, which ones contain meaningful interactions and should be displayed to the end user. In doing so, ACD aims to be informative enough to capture meaningful feature interactions while displaying a sufficiently small subset of all feature groups to maintain simplicity.

## 3 METHOD

This section introduces ACD through two contributions: Sec 3.1 proposes a generalization of CD from LSTMs to arbitrary DNNs, and Sec 3.2 explains the main contribution: how to combine these CD scores with hierarchical clustering to produce ACD.

### 3.1 CONTEXTUAL DECOMPOSITION (CD) IMPORTANCE SCORES FOR GENERAL DNNS

In order to generalize CD to a wider range of DNNs, we first reformulate the original CD algorithm into a more generic setting than originally presented. For a given DNN $f(x)$, we can represent its output as a SoftMax operation applied to logits $g(x)$. These logits, in turn, are the composition of $L$ layers $g_i$, such as convolutional operations or ReLU non-linearities.

$$f(x) = \text{SoftMax}(g(x)) = \text{SoftMax}(g_L(g_{L-1}(...(g_2(g_1(x)))))) \tag{1}$$

Given a group of features $\{x_j\}_{j \in S}$, our generalized CD algorithm, $g^{CD}(x)$, decomposes the logits $g(x)$ into a sum of two terms, $\beta(x)$ and $\gamma(x)$. $\beta(x)$ is the importance measure of the feature group $\{x_j\}_{j \in S}$, and $\gamma(x)$ captures contributions to $g(x)$ not included in $\beta(x)$.

$$g^{CD}(x) = (\beta(x), \gamma(x)) \tag{2}$$
$$\beta(x) + \gamma(x) = g(x) \tag{3}$$

To compute the CD decomposition for $g(x)$, we define layer-wise CD decompositions $g_i^{CD}(x) = (\beta_i, \gamma_i)$ for each layer $g_i(x)$. Here, $\beta_i$ corresponds to the importance measure of $\{x_j\}_{j \in S}$ to layer $i$, and $\gamma_i$ corresponds to the contribution of the rest of the input to layer $i$. To maintain the decomposition we require $\beta_i + \gamma_i = g_i(x)$ for each $i$. We then compute CD scores for the full network by composing these decompositions.

$$g^{CD}(x) = g_L^{CD}(g_{L-1}^{CD}(...(g_2^{CD}(g_1^{CD}(x))))) \tag{4}$$

Previous work (Murdoch et al., 2018) introduced decompositions $g_i^{CD}$ for layers used in LSTMs. The generalized CD described here extends CD to other widely used DNNs, by introducing layer-wise CD decompositions for convolutional, max-pooling, ReLU non-linearity and dropout layers. Doing so generalizes CD scores from LSTMs to a wide range of neural architectures, including CNNs with residual and recurrent architectures.

At first, these decompositions were chosen through an extension of the CD rules detailed in Murdoch et al. (2018), yielding a similar algorithm to that developed concurrently by Godin et al. (2018). However, we found that this algorithm did not perform well on deeper, ImageNet CNNs. We subsequently modified our CD algorithm by partitioning the biases in the convolutional layers between $\gamma_i$ and $\beta_i$ in Equation 5, and modifying the decomposition used for ReLUs in Equation 10. We show the effects of these two changes in Supplement S7, and give additional intuition in Supplement S1.

When $g_i$ is a convolutional or fully connected layer, the layer operation consists of a weight matrix $W$ and a bias $b$. The weight matrix can be multiplied with $\beta_{i-1}$ and $\gamma_{i-1}$ individually, but the bias

must be partitioned between the two. We partition the bias proportionally based on the absolute value of the layer activations. For the convolutional layer, this equation yields only one activation of the output; it must be repeated for each activation.

$$\beta_i = W\beta_{i-1} + \frac{|W\beta_{i-1}|}{|W\beta_{i-1}| + |W\gamma_{i-1}|} \cdot b \tag{5}$$

$$\gamma_i = W\gamma_{i-1} + \frac{|W\gamma_{i-1}|}{|W\beta_{i-1}| + |W\gamma_{i-1}|} \cdot b \tag{6}$$

When $g_i$ is a max-pooling layer, we identify the indices, or channels, selected by max-pool when run by $g_i(x)$, denoted $max\_idxs$ below, and use the decompositions for the corresponding channels.

$$max\_idxs = \underset{idxs}{\operatorname{argmax}} \left[ \operatorname{maxpool}(\beta_{i-1} + \gamma_{i-1}; idxs) \right] \tag{7}$$

$$\beta_i = \beta_{i-1}[max\_idxs] \tag{8}$$

$$\gamma_i = \gamma_{i-1}[max\_idxs] \tag{9}$$

Finally, for the ReLU, we update our importance score $\beta_i$ by computing the activation of $\beta_{i-1}$ alone and then update $\gamma_i$ by subtracting this from the total activation.

$$\beta_i = \operatorname{ReLU}(\beta_{i-1}) \tag{10}$$

$$\gamma_i = \operatorname{ReLU}(\beta_{i-1} + \gamma_{i-1}) - \operatorname{ReLU}(\beta_{i-1}) \tag{11}$$

For a dropout layer, we simply apply dropout to $\beta_{i-1}$ and $\gamma_{i-1}$ individually, or multiplying each by a scalar. Computationally, a CD call is comparable to a forward pass through the network $f$.

## 3.2 Agglomerative Contextual Decomposition (ACD)

Given the generalized CD scores introduced above, we now introduce the clustering procedure used to produce ACD interpretations. At a high-level, our method is equivalent to agglomerative hierarchical clustering, where the CD interaction is used as the joining metric to determine which clusters to join at each step. This procedure builds the hierarchy by starting with individual features and iteratively combining them based on the interaction scores provided by CD. The displayed ACD interpretation is the hierarchy, along with the CD importance score at each node.

More precisely, algorithm 1 describes the exact steps in the clustering procedure. After initializing by computing the CD scores of each feature individually, the algorithm iteratively selects all groups of features within k% of the highest-scoring group (where $k$ is a hyperparameter, fixed at 95 for images and 90 for text) and adds them to the hierarchy.

Each time a new group is added to the hierarchy, a corresponding set of candidate groups is generated by adding individual contiguous features to the original group. For text, the candidate groups correspond to adding one adjacent word onto the current phrase, and for images adding any adjacent pixel onto the current image patch. Candidate groups are ranked according to the CD interaction score, which is the difference between the score of the candidate and original groups.

ACD terminates after an application-specific criterion is met. For sentiment classification, we stop once all words are selected. For images, we stop after some predefined number of iterations and then merge the remaining groups one by one using the same selection criteria described above.

Algorithm 1 is not specific to DNNs; it requires only a method to obtain importance scores for groups of input features. Here, we use CD scores to arrive at the ACD algorithm, which makes the method specific to DNNs, but given a feature group scoring function, Algorithm 1 can yield interpretations for any predictive model. CD is a natural score to use for DNNs as it aggregates saliency at different scales and converges to the final prediction once all the units have been selected.

---

**Algorithm 1** Agglomeration algorithm.

---
**ACD**(Example x, model, hyperparameter k, function CD(x, blob; model))

   # initialize
   tree = Tree()                                                          # tree to output
   scoresQueue = PriorityQueue()                            # scores, sorted by importance
   **for** feature in x :
      scoresQueue.push(feature, priority=CD(x, feature; model))

   # iteratively build up tree
   **while** scoresQueue is not empty :
      selectedGroups = scoresQueue.popTopKPercentile(k)          # pop off top k elements
      tree.add(selectedGroups)               # Add top k elements to the tree

      # generate new groups of features based on current groups and add them to the queue
      **for** selectedGroup in selectedGroups :
         candidateGroups = getCandidateGroups(selectedGroup)
         **for** candidateGroup  in candidateGroups :
            scoresQueue.add(candidateGroup, priority=CD(x, candidateGroup;model)-CD(x,selectedGroup; model))
   **return** tree

---

## 4   RESULTS

We now present empirical validation of ACD on both LSTMs trained on SST and CNNs trained on MNIST and ImageNet. First, we introduce the reader to our visualization in Sec 4.2, and how it can (anecdotally) be used to understand models in settings such as diagnosing incorrect predictions, identifying dataset bias, and identifying representative phrases of differing lengths. We then provide quantitative evidence of the benefits of ACD in Sec 4.3 through human experiments and demonstrating the stability of ACD to adversarial perturbations.

### 4.1   EXPERIMENTAL DETAILS

We first describe the process for training the models from which we produce interpretations. As the objective of this paper is to interpret the predictions of models, rather than increase their predictive accuracy, we use standard best practices to train our models. All models are implemented using PyTorch. For SST, we train a standard binary classification LSTM model[2], which achieves 86.2% accuracy. On MNIST, we use the standard PyTorch example[3], which attains accuracy of 97.7%. On ImageNet, we use a pre-trained VGG-16 DNN architecture Simonyan & Zisserman (2014) which attains top-1 accuracy of 42.8%. When using ACD on ImageNet, for computational reasons, we start the agglomeration process with 14-by-14 superpixels instead of individual pixels. We also smooth the computed image patches by adding pixels surrounded by the patch. The weakened models for the human experiments are constructed from the original models by randomly permuting a small percentage of their weights. For SST/MNIST/ImageNet, 25/25/0.8% of weights are randomized, reducing test accuracy from 85.8/97.7/42.8% to 79.8/79.6/32.3%.

### 4.2   QUALITATIVE EXPERIMENTS

Before providing quantitative evidence of the benefits of ACD, we first introduce the visualization and demonstrate its utility in interpreting a predictive model's behavior. To qualitatively evaluate ACD, in Supplement S3 we show the results of several more examples selected using the same criterion as in our human experiments described below.

#### 4.2.1   UNDERSTANDING PREDICTIVE MODELS USING ACD

In the following examples, we demonstrate the use of ACD to diagnose incorrect predictions in SST and identify dataset bias in ImageNet. These examples are only a few of the potential uses of ACD.

---

[2]model and training code from https://github.com/clairett/pytorch-sentiment-classification
[3]model and training code from https://github.com/pytorch/examples/tree/master/mnist

| Length | Positive | Negative |
|---|---|---|
| 1 | pleasurable, sexy, glorious | nowhere, grotesque, sleep |
| 3 | amazing accomplishment., great fun. | bleak and desperate, conspicuously lacks. |
| 5 | a pretty amazing accomplishment. | ultimately a pointless endeavour. |
| 8 | presents it with an unforgettable visual panache. | my reaction in a word: disappointment. |

Table 1: Top-scoring phrases of different lengths extracted by ACD on SST's validation set. The positive/negative phrases identified by ACD are all indeed positive/negative.

**Text example - diagnosing incorrect predictions**  In the first example, we show the result of running ACD for our SST LSTM model in Figure 2. We can use this ACD visualization to quickly diagnose why the LSTM made an incorrect prediction. In particular, note that the ACD summary of the LSTM correctly identifies two longer phrases and their corresponding sentiment *a great ensemble cast* (positive) and *n't lift this heartfelt enterprise out of the ordinary* (negative). It is only when these two phrases are joined that the LSTM inaccurately predicts a positive sentiment. This suggests that the LSTM has erroneously learned a positive interaction between these two phrases. Prior methods would not be capable of detecting this type of useful information.

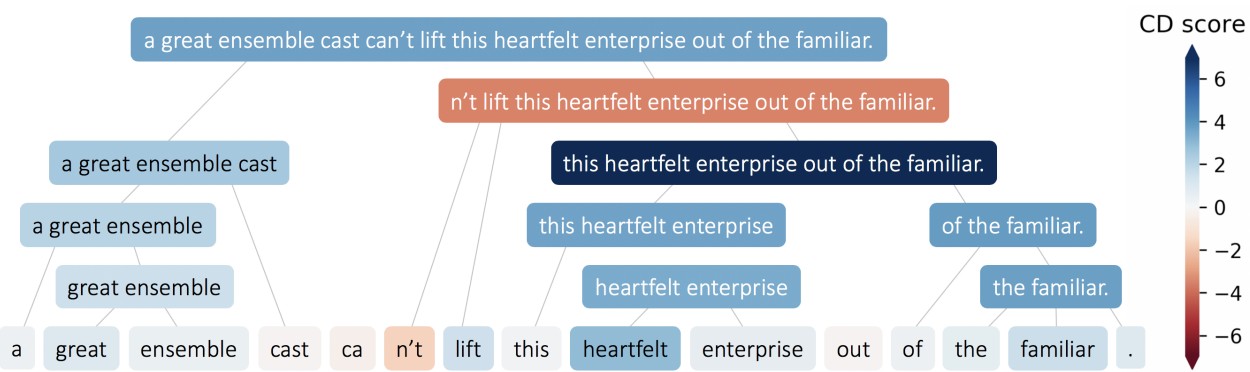

Figure 2: ACD interpretation of an LSTM predicting sentiment. Blue is positive sentiment, white is neutral, red is negative. The bottom row displays CD scores for individual words in the sentence. Higher rows display important phrases identified by ACD, along with their CD scores, converging to the model's (incorrect) prediction in the top row. (Best viewed in color)

**Vision example - identifying dataset bias**  Fig 3 shows an example using ACD for an ImageNet VGG model. Using ACD, we can see that to predict "puck", the CNN is not just focusing on the puck in the image, but also on the hockey player's skates. Moreover, by comparing the fifth and sixth plots in the third row, we can see that the network is only able to distinguish between the class "puck" and the other top classes when the orange skate and green puck patches merge into a single orange patch. This suggests that the CNN has learned that skates are a strong corroborating features for pucks. While intuitively reasonable in the context of ImageNet, this may not be desirable behavior if the model were used in other domains.

### 4.2.2 IDENTIFYING TOP-SCORING PHRASES

When feasible, a common means of scrutinizing what a model has learned is to inspect its most important features, and interactions. In Table 1, we use ACD to show the top-scoring phrases of different lengths for our LSTM trained on SST. These phrases were extracted by running ACD separately on each sample in SST's validation set. The score of each phrase was then computed by averaging over the score it received in each occurrence in a ACD hierarchy. The extracted phrases are clearly reflective of the corresponding sentiment, providing additional evidence that ACD is able to capture meaningful positive and negative phrases. Additional phrases are given in Supplement S2.

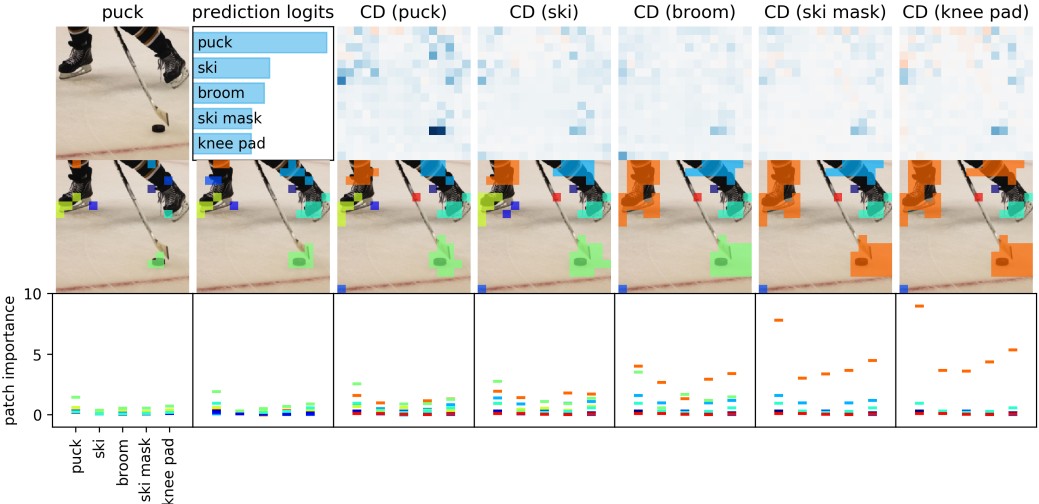

Figure 3: ACD interpretation for a VGG network prediction, described in 4.2.1. ACD shows that the CNN is focusing on skates to predict the class "puck", indicating that the model has captured dataset bias. The top row shows the original image, logits for the five top-predicted classes, and the CD superpixel-level scores for those classes. The second row shows separate image patches ACD has identified as being independently predictive of the class "puck". Starting from the left, each image shows a successive iteration in the agglomeration procedure. The third row shows the CD scores for each of these patches, where patch colors in the second row correspond to line colors in the third row. ACD successfully finds important regions for the target class (such as the puck), and this importance increases as more pixels are selected. Best viewed in color.

## 4.3 QUANTITATIVE EXPERIMENTS

Having introduced our visualization and provided qualitative evidence of its uses, we now provide quantitative evidence of the benefits of ACD.

### 4.3.1 HUMAN EXPERIMENTS

We now demonstrate through human experiments that ACD allows users to better trust and reason about the accuracy of DNNs. Human subjects consist of eleven graduate students at the author's institution, each of whom has taken a class in machine learning. Each subject was asked to fill out a survey with two types of questions: whether, using ACD, they could identify the more accurate of two models and whether they trusted a models output. In both cases, similar questions were asked on three datasets (SST, MNIST and ImageNet), and ACD was compared against three baselines: CD (Murdoch et al., 2018), Integrated Gradients (IG) (Sundararajan et al., 2017), and occlusion (Li et al., 2016; Zeiler & Fergus, 2014). The exact survey prompts are provided in Supplement S4.

**Identifying an accurate model** The objective of this section was to determine if subjects could use a small number of interpretations produced by ACD in order to identify the more accurate of two models. For each question in this section, two example predictions were chosen. For each of these two predictions, subjects were given interpretations from two different models (four total), and asked to identify which of the two models had a higher predictive accuracy. Each subject was asked to make this comparison using three different sets of examples for each combination of dataset and interpretation method, for 36 total comparisons. To remove variance due to examples, the same three sets of examples were used across all four interpretation methods.

The predictions shown were chosen to maximize disagreement between models, with SST also being restricted to sentences between five and twenty words, for ease of visualization. To prevent subjects from simply picking the model that predicts more accurately for the given example, for each question a user is shown two examples: one where only the first model predicts correctly and one where only

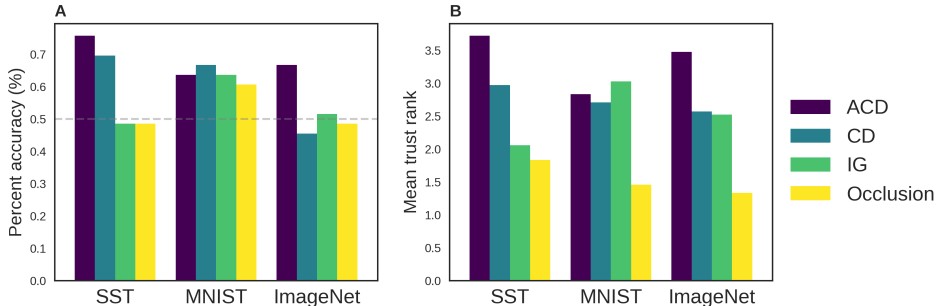

Figure 4: Results for human studies. **A.** Binary accuracy for whether a subject correctly selected the more accurate model using different interpretation techniques **B.** Average rank (from 1 to 4) of how much different interpretation techniques helped a subject to trust a model, higher ranks are better.

the second model predicts correctly. The two models considered were the accurate models of the previous section and a weakened version of that same model (details given in Sec 4.1).

Fig 4A shows the results of the survey. For SST, humans were better able to identify the strongly predictive model using ACD compared to other baselines, with only ACD and CD outperforming random selection (50%). Based on a one-sided two-sample t-test, the gaps between ACD and IG/Occlusion are significant, but not the gap between ACD and CD. In the simple setting of MNIST, ACD performs similarly to other methods. When applied to ImageNet, a more complex dataset, ACD substantially outperforms prior, non-hierarchical methods, and is the only method to outperform random chance, although the gaps between ACD and other methods are only statistically suggestive (p-values fall between 0.15 and 0.07).

**Evaluating trust in a model** In this section, the goal is to gauge whether ACD helps a subject to better trust a model's predictions, relative to prior techniques. For each question, subjects were shown interpretations of the same prediction using four different interpretation methods, and were asked to rank the interpretations from one to four based on how much they instilled trust in trust the model. Subjects were asked to do this ranking for three different examples in each dataset, for nine total rankings. The interpretations were produced from the more accurate model from the previous section, and the examples were chosen using the same criteria as the previous section, except they were restricted to examples correctly predicted by the more accurate model.

Fig 4B shows the average ranking received by each method/dataset pair. ACD substantially outperforms other baselines, particularly for ImageNet, achieving an average rank of 3.5 out of 4, where higher ranks are better. As in the prior question, we found that the hierarchy only provided benefits in the more complicated ImageNet setting, with results on MNIST inconclusive. For both SST and ImageNet, the difference in mean ranks between ACD and all other methods is statistically significant (p-value less than 0.005) based on a permutation test, while on MNIST only the difference between ACD and occlusion is significant.

### 4.3.2 ACD HIERARCHY IS ROBUST TO ADVERSARIAL PERTURBATIONS

While there has been a considerable amount of work on adversarial attacks, little effort has been devoted to qualitatively understanding this phenomenon. In this section, we provide evidence that, on MNIST, the hierarchical clustering produced by ACD is largely robust to adversarial perturbations. This suggests that ACD's hierarchy captures fundamental features of an image, and is largely immune to the spurious noise favored by adversarial examples.

To measure the robustness of ACD's hierarchy, we first qualitatively compare the interpretations produced by ACD on both an unaltered image and an adversarially perturbed version of that image. Empirically, we found that the extracted hierarchies are often very similar, see Supplement S5. To generalize these observations, we introduce a metric to quantify the similarity between two ACD hierarchies. This metric allows us to make quantitative, dataset-level statements about the stability of ACD feature hierarchies with respect to adversarial inputs. Given an ACD hierarchy, we com-

| Attack Type | ACD | Agglomerative Occlusion |
|---|---|---|
| Saliency (Papernot et al., 2016) | 0.762 | 0.259 |
| Gradient attack | 0.662 | 0.196 |
| FGSM (Goodfellow et al., 2014) | 0.590 | 0.131 |
| Boundary (Brendel et al., 2017) | 0.684 | 0.155 |
| DeepFool (Moosavi Dezfooli et al., 2016) | 0.694 | 0.202 |

Table 2: Correlation between pixel ranks for different adversarial attacks. ACD achieves consistently high correlation across different attack types, indicating that ACD hierarchies are largely robust to adversarial attacks. Using occlusion in place of CD produces substantially less stable hierarchies.

pute a ranking of the input image's pixels according to the order in which they were added to the hierarchy. To measure the similarity between the ACD hierarchies for original and adversarial images, we compute the correlation between their corresponding rankings. As ACD hierarchies are class-specific, we average the correlations for the original and adversarially altered predictions.

We display the correlations for five different attacks (computed using the Foolbox package Rauber et al. (2017), examples shown in Supplement S6), each averaged over 100 randomly chosen predictions, in Table 2. As ACD is the first local interpretation technique to compute a hierarchy, there is little prior work available for comparison. As a baseline, we use our agglomeration algorithm with occlusion in place of CD. The resulting correlations are substantially lower, indicating that features detected by ACD are more stable to adversarial attacks than comparable methods. These results provide evidence that ACD's hierarchy captures fundamental features of an image, and is largely immune to the spurious noise favored by adversarial examples.

## 5 CONCLUSION

In this work, we introduce agglomerative contextual decomposition (ACD), a novel hierarchical interpretation algorithm. ACD is the first method to use a hierarchy to interpret individual neural network predictions. Doing so enables ACD to automatically detect and display non-linear contributions to individual DNN predictions, something prior interpretation methods are unable to do. The benefits of capturing the non-linearities inherent in DNNs are demonstrated through human experiments and examples of diagnosing incorrect predictions and dataset bias. We also demonstrate that ACD's hierarchy is robust to adversarial perturbations in CNNs, implying that it captures fundamental aspects of the input and ignores spurious noise.

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

# ACD SUPPLEMENT

## S1   CD SCORE COMPARISONS

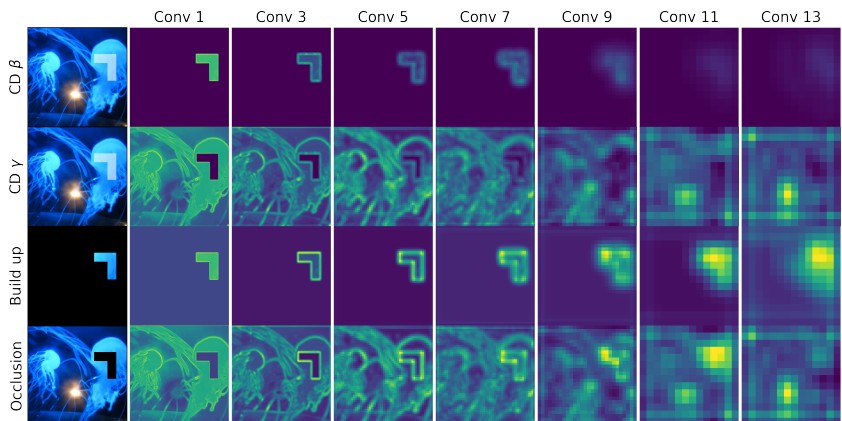

Figure S1: Intuition for CD run on a corner-shaped blob compared to *build-up* and *occlusion*. CD decomposes a DNN's feedforward pass into a part from the blob of interest (top row) and everything else (second row). Left column shows original image with overlaid blob. Other columns show DNN activations summed over the filter dimension. Top and third rows are on same color scale. Second and bottom rows ar[...]

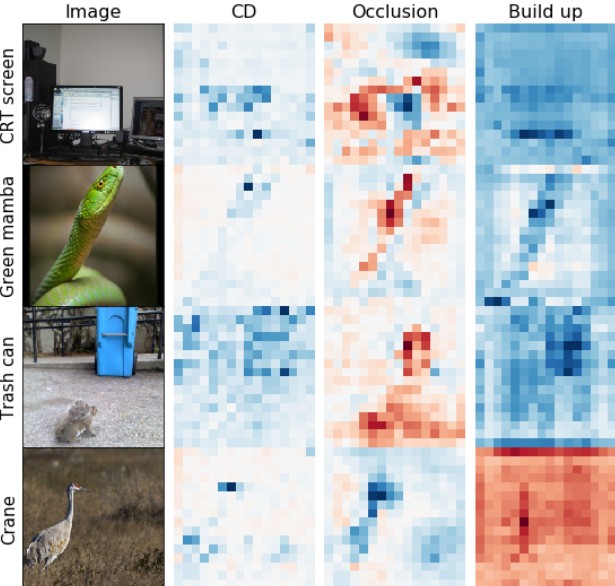

Figure S2: Comparing unit-level CD scores for the correct class to scores from baseline methods. In each case, the model correctly predicts the label, shown on the y axis. Blue is positive, white is neutral, and red is negative. Best viewed in color.

Fig S1 gives intuition for CD on the VGG-16 ImageNet model described in Sec 4. CD keeps track of the contributions of the blob and non-blob throughout the network. This is intuitively similar to the occlusion and build-up methods, shown in the bottom two rows. The build-up method sets everything but the patch of interest to a references value (often zero). These rows compare the CD decomposition to perturbing the input as in the occlusion and build-up methods. They are similar in early layers, but differences become apparent in later layers.

| Length | Positive | Negative |
|---|---|---|
| 1 | 'pleasurable', 'sexy', 'glorious', 'delight', 'unforgettable' | 'nowhere', 'grotesque', 'sleep', 'mundane', 'clich' |
| 3 | 'amazing accomplishment .', 'great fun .', 'good fun .', 'language sexy .', 'are magnificent .' | 'very bad .', ': disappointment .', 'quite bad .', 'conspicuously lacks .', 'bleak and desperate' |
| 5 | 'a pretty amazing accomplishment .', 'clearly , great fun .', 'richness of its performances .', 'a delightful coming-of-age story .', 'an unforgettable visual panache .' | 'ultimately a pointless endeavor .', 'this is so bad .', 'emotion closer to pity .', 'fat waste of time .', 'sketch gone horribly wrong .' |
| 8 | 'presents it with an unforgettable visual panache .', 'film is packed with information and impressions .', 'entertains by providing good , lively company .' | 'my reaction in a word : disappointment .', ''s slow – very , very slow .', 'a dull , ridiculous attempt at heart-tugging .' |
| 12 | 'in delicious colors , and the costumes and sets are grand .', 'part stevens glides through on some solid performances and witty dialogue .', 'mamet enthusiast and for anyone who appreciates intelligent , stylish moviemaking .' | ''actors provide scant reason to care in this crude '70s throwback .', 'more often just feels generic , derivative and done to death .', 'its storyline with glitches casual fans could correct in their sleep .' |
| 15 | 'serry shows a remarkable gift for storytelling with this moving , effective little film .', ', lathan and diggs are charming and have chemistry both as friends and lovers .' | 'level that one enjoys a bad slasher flick , primarily because it is dull .', 'technicality that strains credulity and leaves the viewer haunted by the waste of potential .' |

Table S1: Top-scoring phrases of different lengths extracted by ACD on SST's validation set. The positive/negative phrases identified by ACD are all indeed positive/negative

Fig S2 compares the 7x7 superpixel-level scores for four images comparing different methods for obtaining importance scores. CD scores better find information relevant to predicting the correct class.

## S2 TOP SCORING ACD PHRASES

Here we provide an extended version of Table S1, containing the top 5 phrases of each length for positive/negative polarities. These were extracted using ACD from an LSTM trained on SST.

## S3 ACD EXAMPLES

We provide additional, automatically selected, visualizations produced by ACD. These examples were chosen using the same criteria as the human experiments describes in Sec 4.3.1. All examples are best viewed in color.

**SST top-predicted examples.** Here, the model used and figure produced correspond to Fig 2.

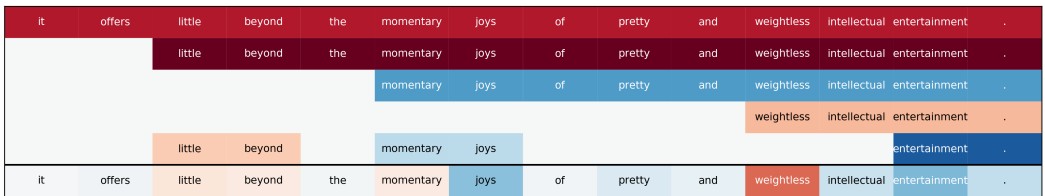

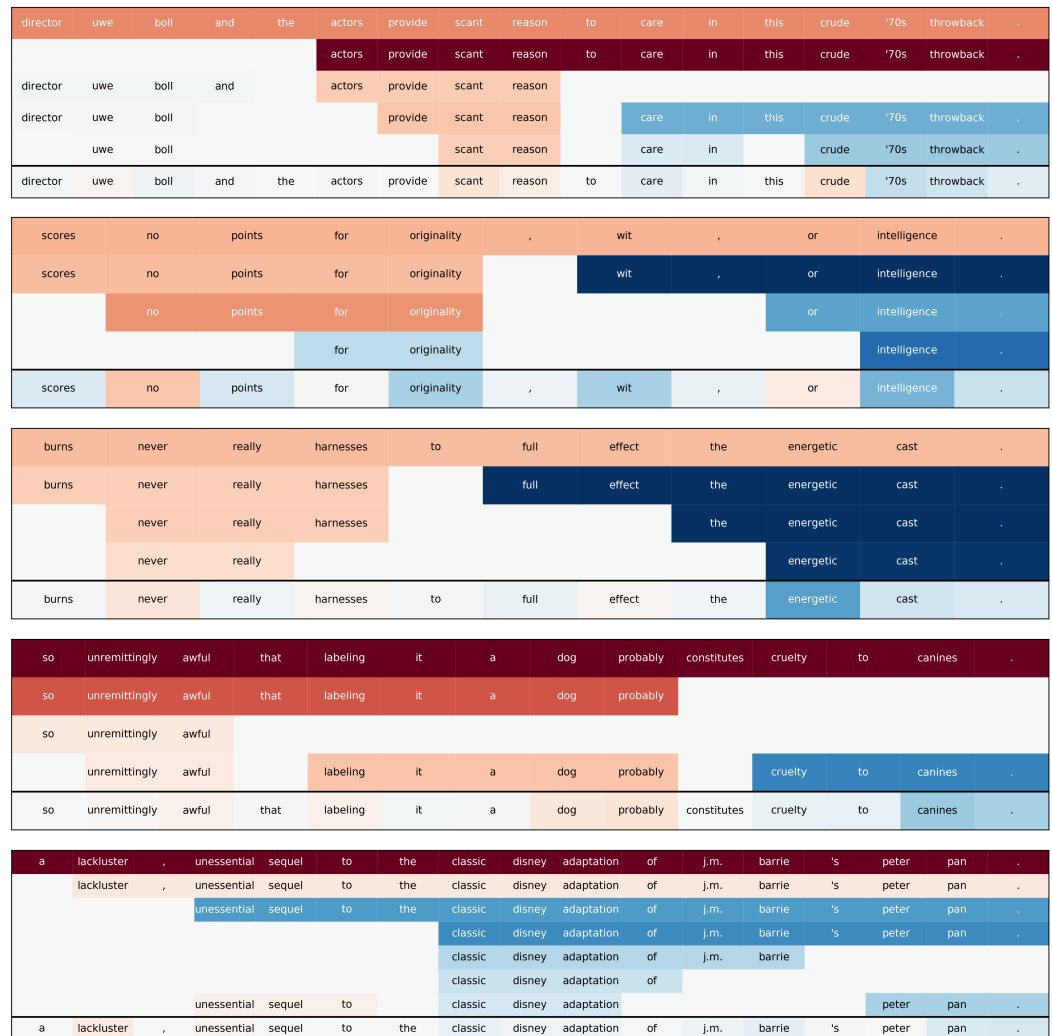

**SST lowest-predicted examples.** Here, the model used and figure produced correspond to Fig 2.

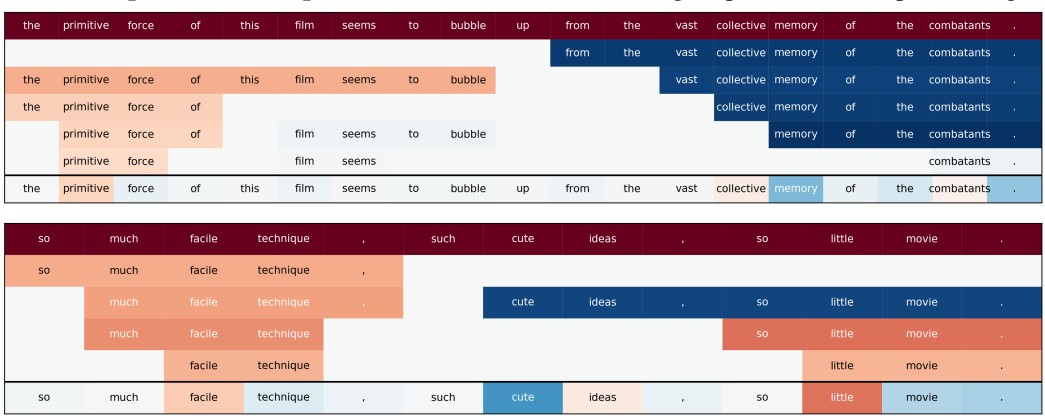

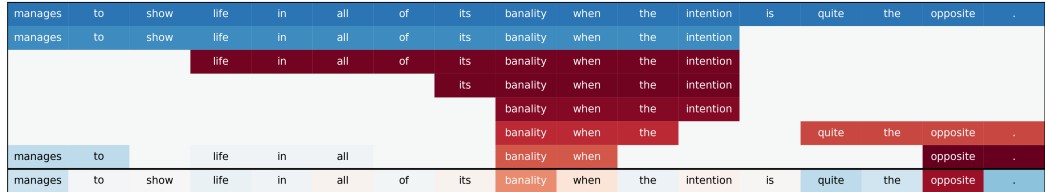

**MNIST top-predicted examples.** Here, the model used is the same as in Sec 4.3.2 and the interpretation of the figure produced is the same as in Fig 3.

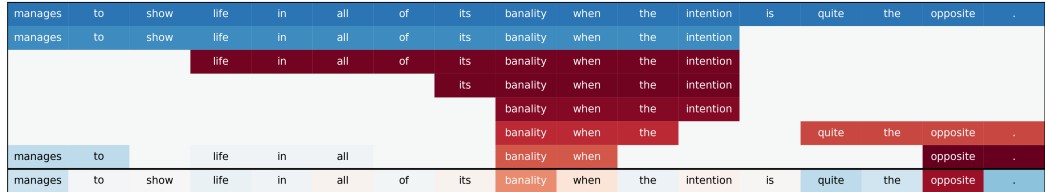

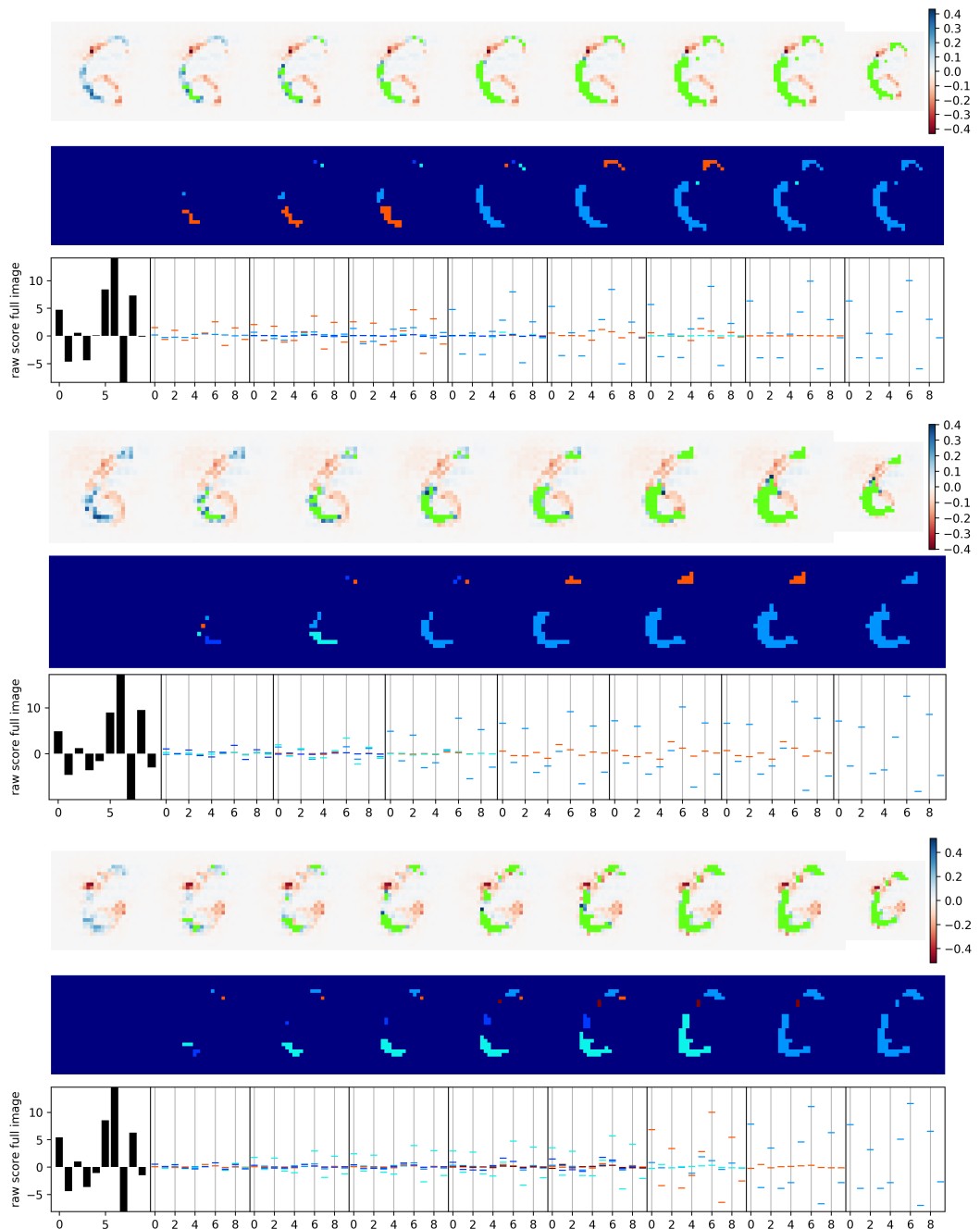

**MNIST lowest-predicted examples.** Here, the model used is the same as in Sec 4.3.2 and the interpretation of the figure produced is the same as in Fig 3.

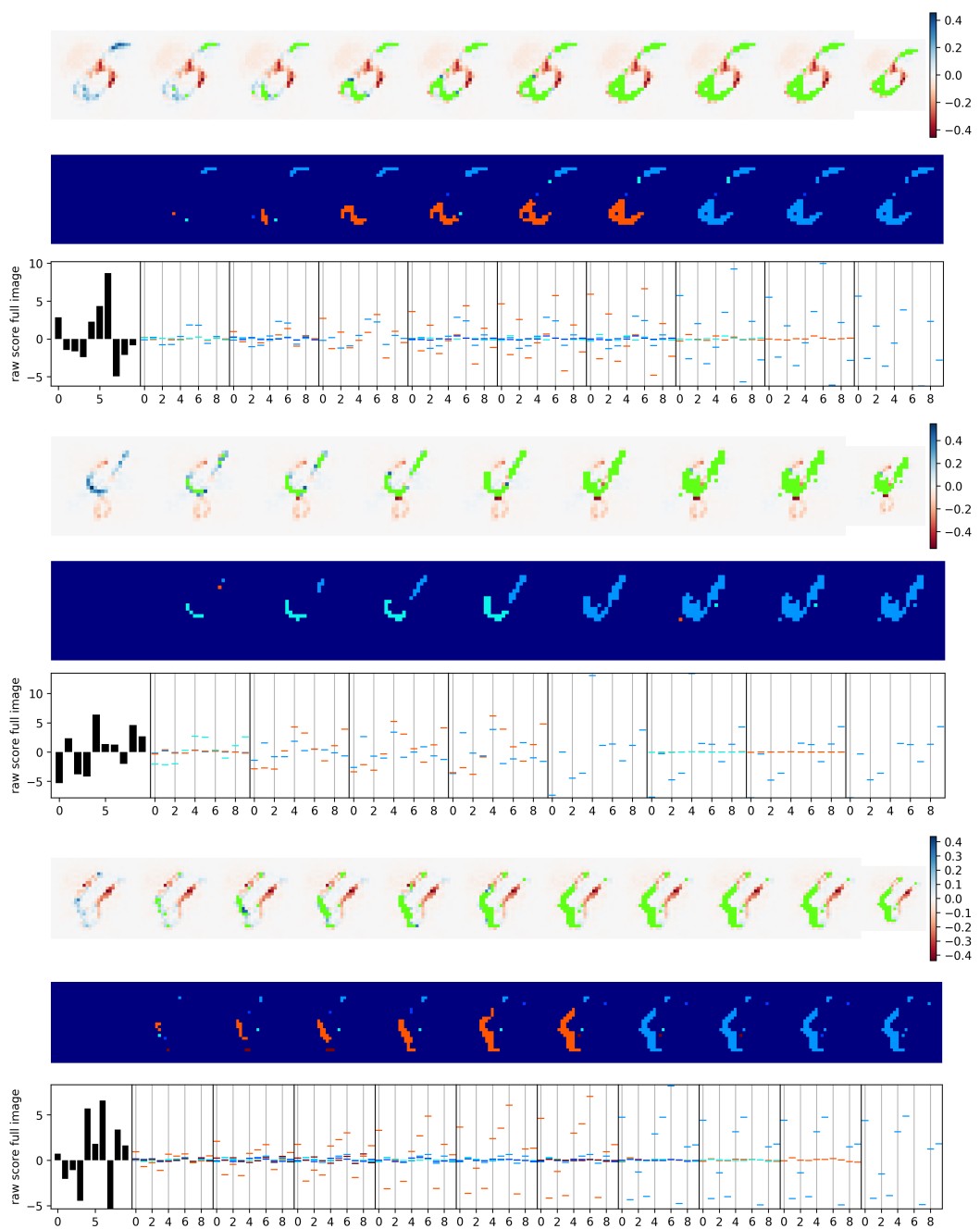

**Imagenet top-predicted examples.** Here, the model used and figure produced correspond to that in Fig 3.

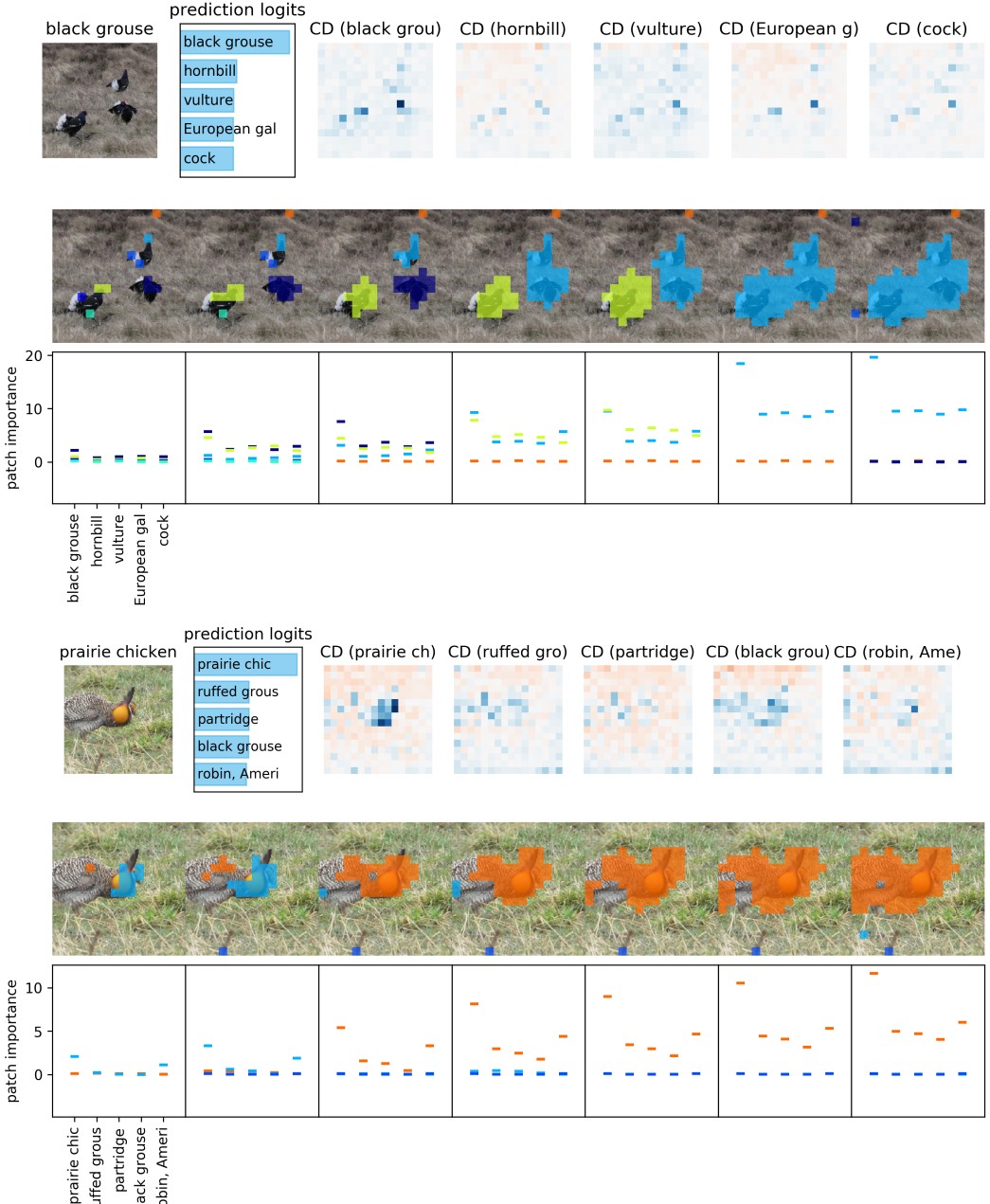

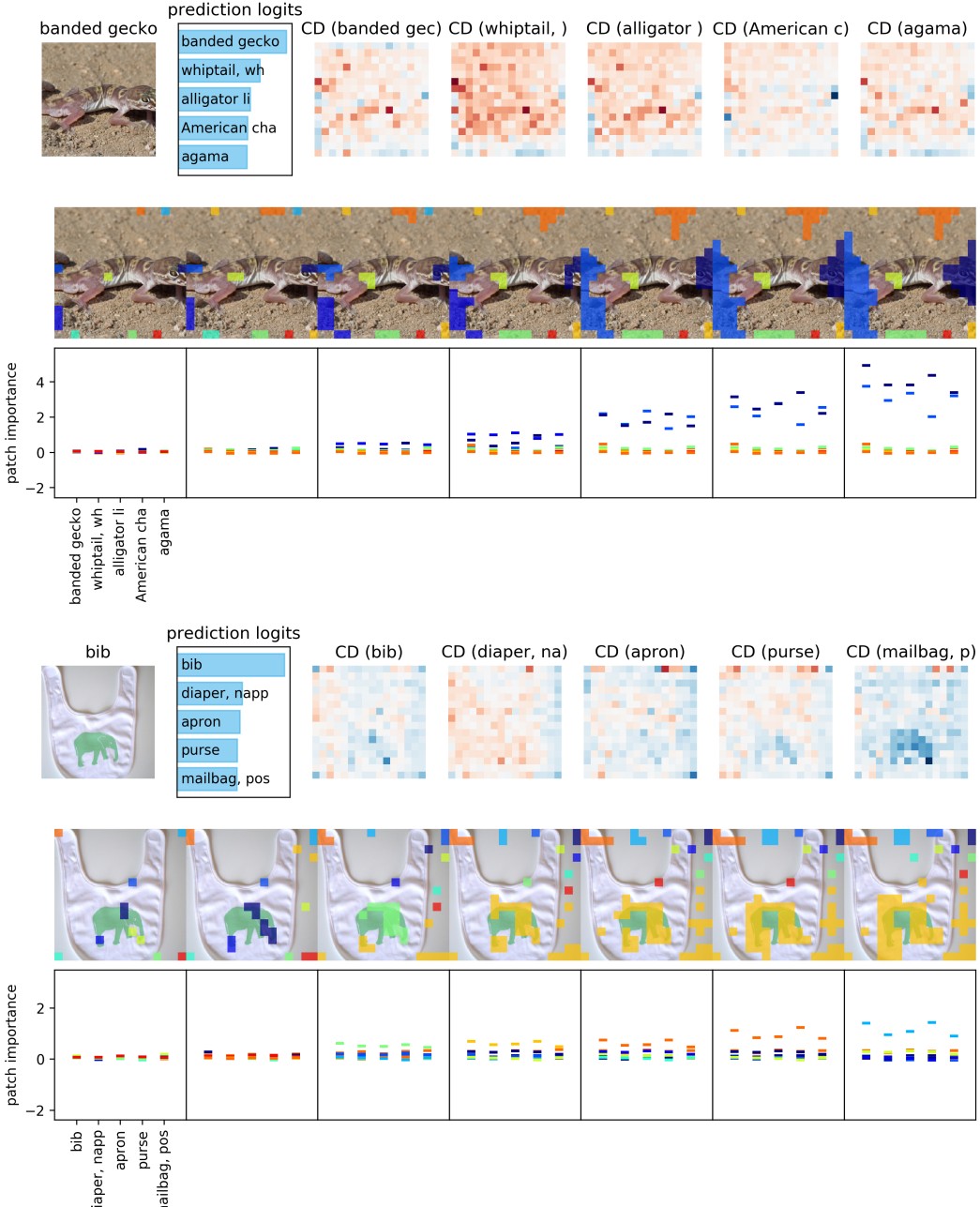

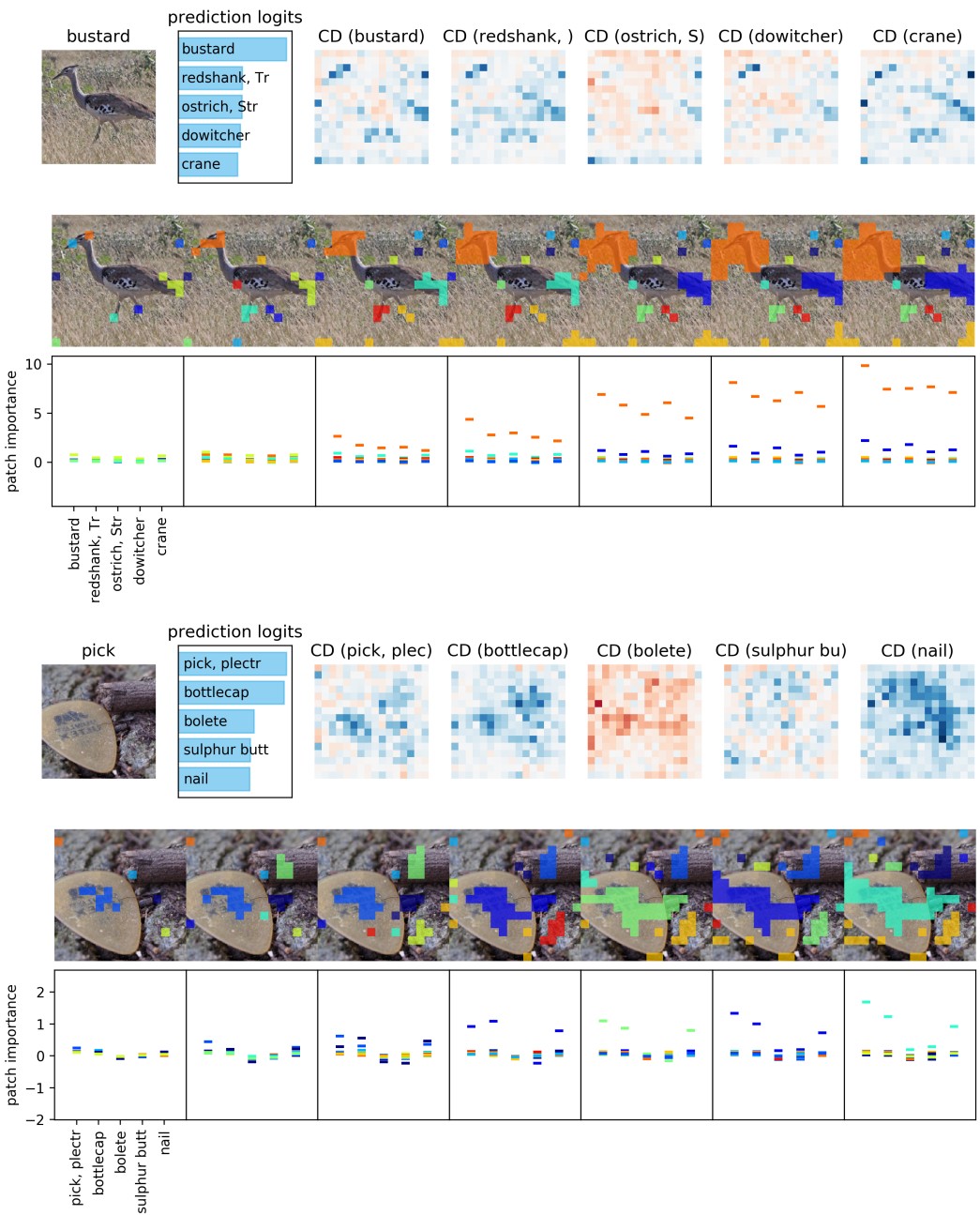

**Imagenet lowest-predicted examples.** Here, the model used and figure produced correspond to that in Fig 3.

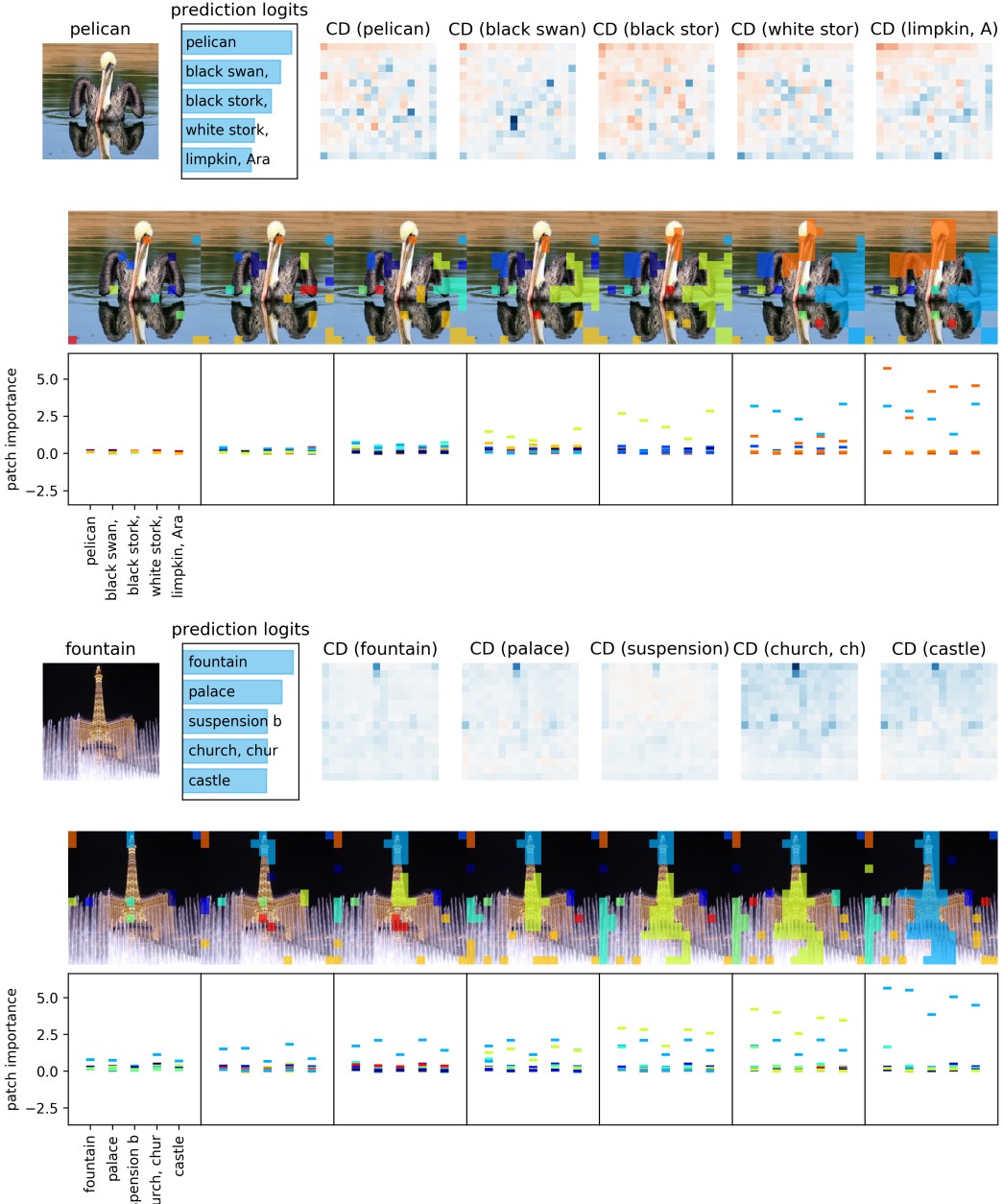

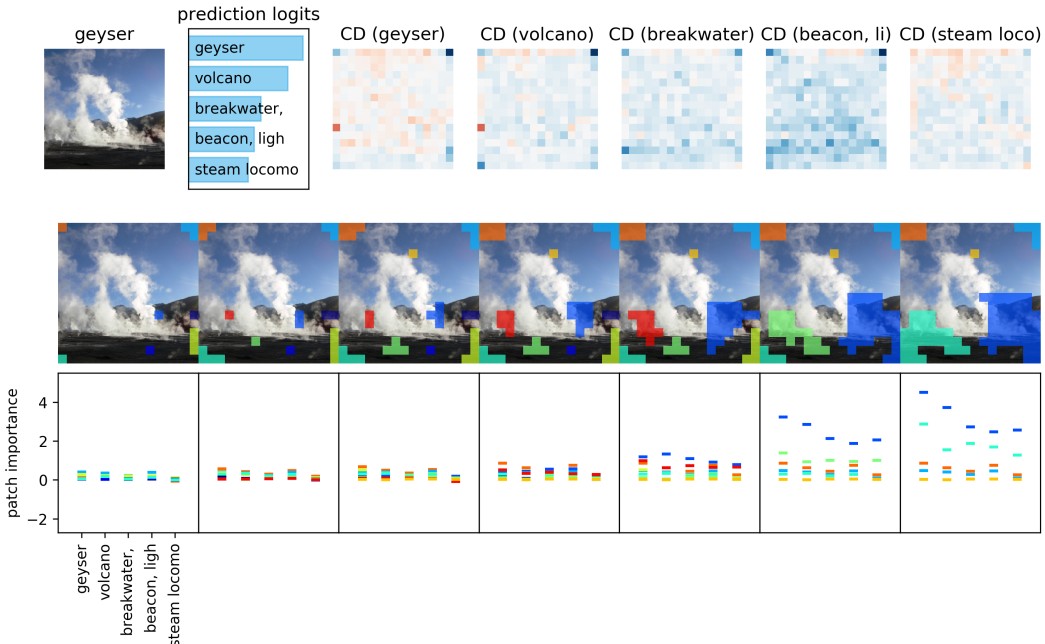

## S4    HUMAN EXPERIMENTS EXPERIMENTAL SETUP

Order of questions is randomized for each subject. Below are the instructions and questions given to the user (for brevity, the actual visualizations are omitted, but are similar to the visualizations shown in Supplement S3).

This survey aims to compare different interpretation techniques. In what follows, blue is positive, white is neutral, and red is negative.

### S4.1    SENTIMENT CLASSIFICATION

#### S4.1.1    CHOOSING THE BETTER MODEL

In this section, the task is to compare two models that classify movie reviews as either positive (good movie) or negative (bad movie). One model has better predictive accuracy than the other.

In what follows, you will see visualizations of what both models have learned. These visualizations use different methods of identifying contributions to the final prediction of either individual words or groups of them. For each model, we show visualizations of two different examples.

In these visualizations, the color shows what the model thinks for individual words / groups of words. Blue is positive sentiment (e.g. "great", "fantastic") and red is negative sentiment (e.g. "terrible", "miserable").

**Using these visualizations, please write A or B to select which model you think has higher predictive accuracy.**

#### S4.1.2    GAUGING TRUST

Now, we show results only from the good model. Your task is to compare different visualizations. For the following predictions, please select which visualization method leads you to trust the model the most.

**Put a number next to each of the following letters ranking them in the order of how much they make you trust the model (1-4, 1 is the most trustworthy).**

### S4.2 MNIST

#### S4.2.1 CHOOSING THE BETTER MODEL

Now we will perform a similar challenge for vision. Your task is to compare two models that classify images into classes, in this case digits from 0-9. One model has higher predictive accuracy than the other.

In what follows, you will see visualizations of what both models have learned. These visualizations use different methods of identifying contributions to the final prediction of either individual pixels or groups of them. Using these visualizations, please select the model you think has higher accuracy.

For each prediction, the top row contains the raw image followed by five heat maps, and the title shows the predicted class. Each heatmap corresponds to a different class, with blue pixels indicating a pixel is a positive signal for that class, and red pixels indicating a negative signal. **The first heatmap title shows the predicted class of the network - this is wrong half the time. In some cases, each visualization has an extra row, which shows groups of pixels**, at multiple levels of granularity, that contribute to the predicted class.

**Using these visualizations, please select which model you think has higher predictive accuracy, A or B.**

#### S4.2.2 GAUGING TRUST

Now, we show results only from the good model. Your task is to compare different visualizations. For the following predictions, please select which visualization method leads you to trust the model the most.

**Put a number next to each of the following letters ranking them in the order of how much they make you trust the model (1-4, 1 is the most trustworthy).**

#### S4.2.3 CHOOSING THE MORE ACCURATE MODEL

Now we will perform a similar challenge for vision. Your task is to compare two models that classify images into classes (ex. balloon, bee, pomegranate). One model is better than the other in terms of predictive accuracy.

In what follows, you will see visualizations of what both models have learned. These visualizations use different methods of identifying contributions to the final prediction of either individual pixels or groups of them.

For each prediction, the top row contains the raw image followed by five heat maps, and the title shows the predicted class. Each heatmap corresponds to a different class, with blue pixels indicating a pixel is a positive signal for that class, and red pixels indicating a negative signal. **The first heatmap title shows the predicted class of the network - this is wrong half the time. In some cases, each visualization has an extra row, which shows groups of pixels**, at multiple levels of granularity, that contribute to the predicted class.

**Using these visualizations, please select which model you think has higher predictive accuracy, A or B.**

#### S4.2.4 GAUGING TRUST

Now, we show results only from the more accurate model. Your task is to compare different visualizations. For the following predictions, please select which visualization method leads you to trust the model's decision the most.

**Put a number next to each of the following letters ranking them in the order of how much they make you trust the model (1-4, 1 is the most trustworthy).**

## S5  ACD ON ADVERSARIAL EXAMPLES

The hierarchies constructed by ACD to explain a prediction of 0 are substantially similar for both the original image and an adversarially perturbed image predicted to be a 6.

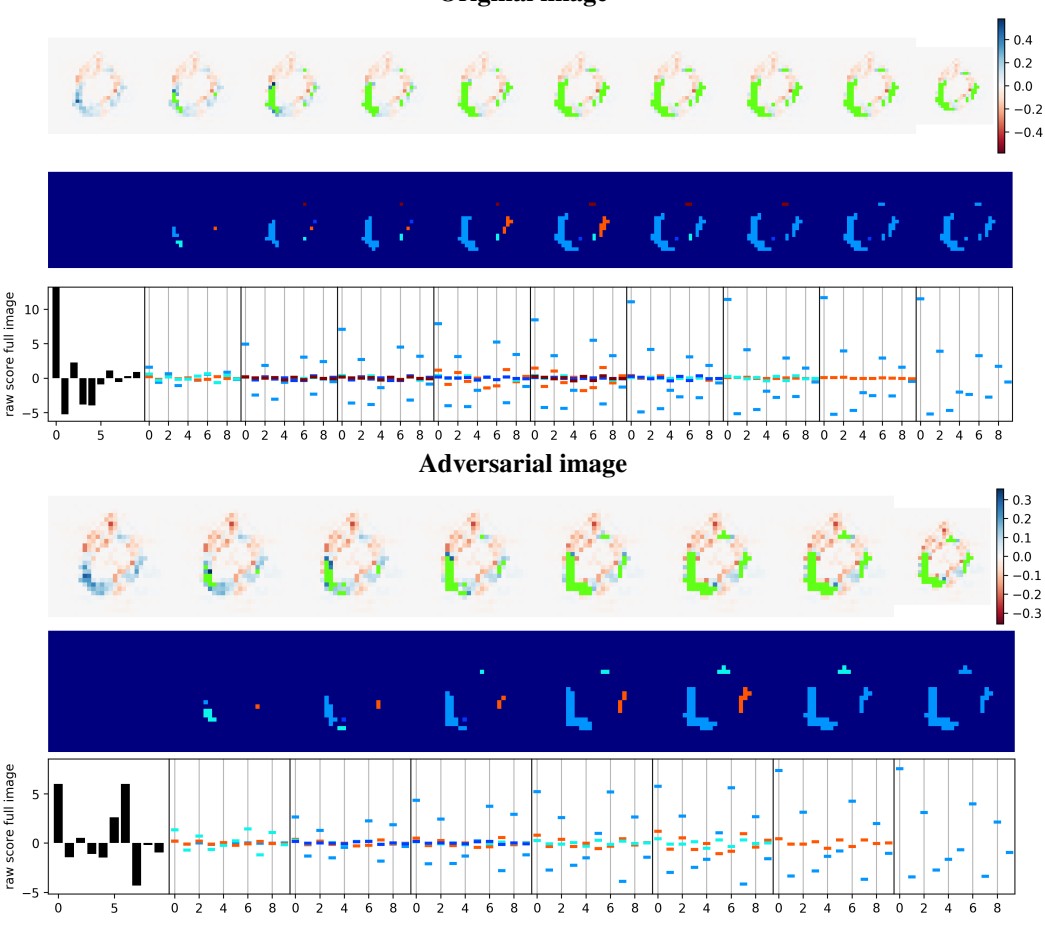

Figure S3: Example of ACD run on an image of class 0 before and after an adversarial perturbation (a DeepFool attack). Best viewed in color.

## S6  ADVERSARIAL ATTACK EXAMPLES

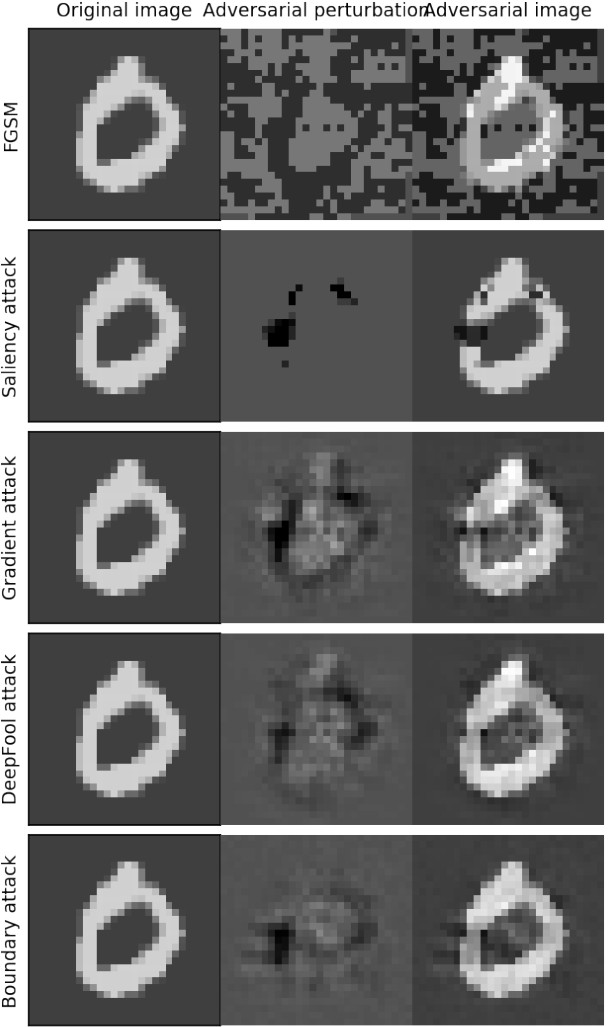

Figure S4: Examples of attacks for one image. Original image (left column) is correctly predicted as class 0. After each adversarial perturbation (middle column), the predicted class for the adversarial image (right column) is now altered.

## S7  GENERALIZING CD TO CNNS

Fig S5 qualitatively shows the change in behavior as the result of two modifications made to the naive extension of CD to CNNs, which was independently developed by Godin et al. (2018). During development of our general CD, two changes were made. First, we partitioned the bias between $\gamma_i$ and $\beta_i$, as described in Equation 5. As can be seen in the second column, this qualitatively reduces the noise in the heat maps. Next, we replace the ReLU Shapely decomposition by the decomposition provided in Equation 10. In the third column, you can see that this effectively prevents the CD scores from becoming unrealistically large in areas that should not be influencing the model's decision. When these two approaches are combined in the fourth column, they provide qualitatively sensible heatmaps with reasonably valued CD scores. When applied to the smaller models used on SST and MNIST, these changes don't have large effects on the interpretations.

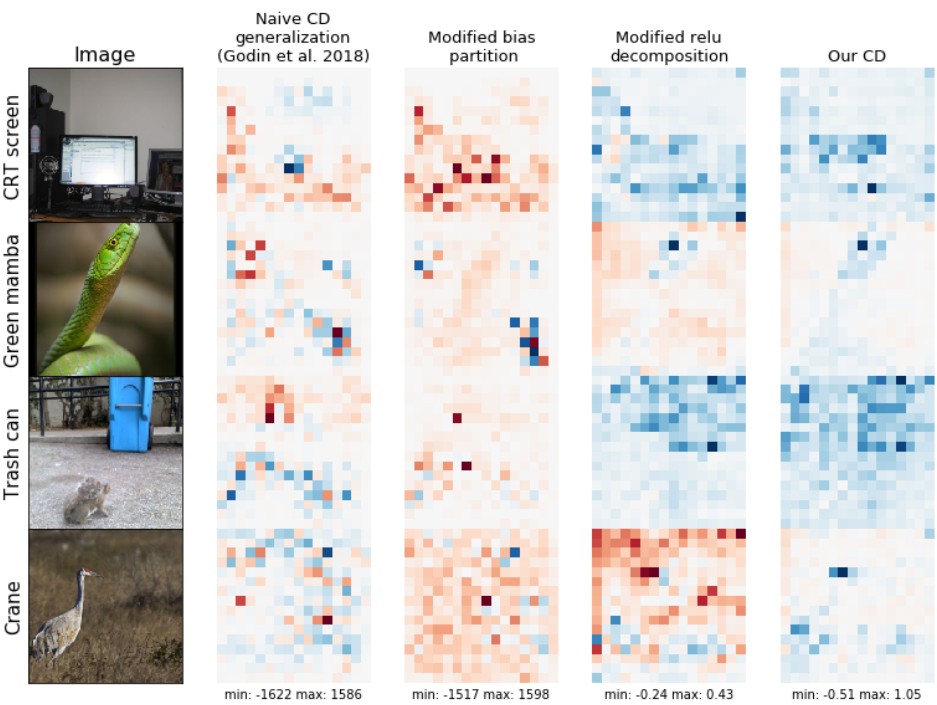

Figure S5: Comparing unit-level CD scores to CD scores from the naive extension of CD to CNNs, independently developed by Godin et al. (2018). Labels under the bottom row signify the minimum and maximum scores from each column. Altering the bias partition and ReLU decomposition qualitatively improves scores (e.g. see scores in bottom row corresponding to the location of the crane), and avoids extremely large magnitudes (see values under left two columns). Blue is positive, white is neutral, and red is negative. In each case, scores are for the correct class, which the model predicts correctly (shown on the y axis).

