# OpenReview forum: "Hierarchical interpretations for neural network predictions"
_ICLR.cc/2019/Conference_

### Official Review · AnonReviewer3 · 2018-10-22
**Interesting idea. Potentially convincing experiments. Limited methodological novelty**

**Rating:** 6
**Confidence:** 4

**Review:**

This paper proposes a novel approach to explain neural network predictions by learning hierarchical representations of groups of input features and their contribution to the final prediction. The proposed method is a straightforward extension of the contextual decomposition work by (Murdoch et. al. 2018) which estimates feature interpretability for LSTMs. This work extends (Murdoch et. al. '18) to more general NN architectures and further employs agglomerative clustering to identify groups of features-- as opposed to individual features--that are predictive of the output.

Results are shown using a LSTM trained on the standard Stanford sentiment task and a VCG DNN trained on ImageNet which show the superior performance of the proposed approach. In addition, the paper also provides some survey results where "humans" were asked to pick more interpretable models.

The paper is nicely written and puts itself nicely in context of the previous work. Though, I have several concerns:

1). Biggest concern: Conditioning on the (Murdoch et. al. 18) paper, the methodological novelty of the proposed approach is minimal. Though, the experimental gains on the vision and NLP tasks are nice.

2). It was unclear to me how the agglomerative algorithm (Algorithm 1) was run. That is, was it run as part of the LSTM estimation for instance for the sentiment task OR was it run post-hoc after getting the model estimates from LSTM? If it was run post-hoc then I am unsure if we can assume that the "agglomeratively grouped CD scores of individual features" are the same as the "CD scores for the groups/interactions of features" in terms of their contribution to the final prediction.

3). Though, the paper mentions several times regarding generalizing (Murdoch et. al. 18) to architectures other than LSTMs but still the experimental results on the sentiment task uses an LSTM as the model. It would have been nice to show the comparative strength of the proposed approach on a different architecture even for the sentiment task. (I understand that the paper uses a different DNN architecture for the vision task).

4). The paper talks several times about diagnosing why a model went wrong e.g. the "negation" in the case of the LSTM model in Figure 2, but never discusses the bigger and more interesting problem. How can we build an improved LSTM model for the sentiment task which classifies that incorrect prediction correctly?

---

> ### Author Response · Authors · 2018-11-08
> **Thanks for your comments**
>
> Thank you very much for the exceptionally detailed and thoughtful review. We address your concerns below
>
> 1) We’d like to emphasize that our main source of novelty lies in moving from the word/pixel-level heat maps (e.g. Figure 5 in [1]), which are the current SOTA for interpreting individual DNN predictions, to hierarchical interpretations, such as Figure 2 in our paper. In our human experiments, we compare our hierarchical interpretations against three non-hierarchical baselines, ultimately concluding that the hierarchy is in fact beneficial.
>
> Given that the experimental gains across NLP and vision are meaningful, we feel that the simplicity of our method should be an argument for its acceptance, not against. In machine learning, there is often the temptation to propose complicated “highly novel” approaches, many of which are never adopted. In contrast, we explicitly tried to identify the simplest approach that could produce a meaningful improvement over the current SOTA. In interpretability, we feel simplicity is particularly important, as we must consider the human component in providing simple, easy to understand insights into the model. We have found that the more complicated the interpretation method, the harder it is to convince users to use, understand, and trust the method’s output.
>
> 2) As R2 also pointed out, section 3.2 was too heavy with mathematical details and lacking intuition. We have now added a paragraph to the beginning of section 3.2, which should hopefully make it clearer.
>
> To answer your question, the agglomeration algorithm was run post-hoc - ACD does not modify the original prediction of the LSTM.
>
> The agglomeration algorithm is simply an approach for constructing the hierarchy of phrases/pixels, but does not alter the CD scores produced for each node in the hierarchy. That is, the score for each node in the hierarchy is simply the CD score for that particular phrase/pixel-blob, which I believe is what you mean by “CD scores for the groups/interactions of features". By "agglomeratively grouped CD scores of individual features", I think you’re referring to the sum of the CD scores for the sub-phrases/words contained within a phrase. This value isn’t displayed in the hierarchy, as summing importance scores can’t capture interactions, such as the negation between “n’t lift” and “this heartfelt enterprise out of the familiar.” that occurs in Figure 2.
>
> 3) We agree that it could be interesting to see whether CNNs and LSTMs produce similar importance scores and/or hierarchies on the same dataset, and this is something we have thought about for future work (e.g. “Do LSTMs and CNNs capture different kinds of interactions?”). Unfortunately, in a conference paper we don’t feel we have the space to do such an analysis in a defensible manner. Moreover, we feel the existing results are more important to justifying the main part of the paper - hierarchical interpretations.
>
> 4) The problem of using interpretations to improve accuracy is quite interesting, and one that we have spent a lot of time thinking about it. The short answer is that it is not immediately clear how to do so, but we are optimistic that improved interpretations like ACD should prove useful in improving model’s accuracy
>
> However, even if ACD never leads to increased prediction performance, we think it’s important to stress that there are many uses for interpretations that have no effect on predictive performance. As we discuss in the first paragraph of our introduction, in scientific applications [2-4], it is the interpretations themselves which are the findings, and are ultimately reported in publications. In industry, interpretability is important in determining the fairness of a model [5], and satisfying regulatory concerns [6]. Finally, as we show in our human experiments, improved interpretations help users to better trust the predictions of their models, which is helpful even if it does not change the predictions themselves.
>
> [1] https://arxiv.org/pdf/1612.08220.pdf
> [2] https://arxiv.org/abs/1702.05747
> [3]https://www.researchgate.net/publication/261538344_The_Emergence_of_Machine_Learning_Techniques_in_Criminology
> [4] http://msb.embopress.org/content/12/7/878
> [5] https://arxiv.org/abs/1104.3913
> [6] https://arxiv.org/abs/1606.08813

---

> > ### Comment · AnonReviewer3 · 2018-11-21
> > **The revised version of the paper is improved.**
> >
> > Thanks for updating the paper. The revised version is definitely improved. Also, thanks for clarifying some of my questions.
> >
> >
> > Regarding your comment:
> > >>>we feel that the simplicity of our method should be an argument for its acceptance, not against.
> >
> > I totally agree with that and am myself a big fan of simpler methods which show strong empirical performance!
> >
> > However, the issue here is a little different. It's not so much about simplicity but more about the incremental contribution of the paper relative to the CD paper by (Murdoch et. al. 18), which in my eyes is little.
> >
> > That said, the judgement of novelty of a paper w.r.t. prior literature is very subjective, so I leave it to the Area Chair/Senior PC to make a final call on that.

---

> > > ### Author Response · Authors · 2018-11-24
> > > **Thanks for responding**
> > >
> > > Thanks for your response, and acknowledging the paper updates.
> > >
> > > To summarize, it appears our revisions have satisfied most of your concerns, with the only remaining one being the "incremental contribution of the paper".  Moreover, this concern is strong enough that you have not altered your rating despite your other concerns being addressed. We have two points we'd like to make in response.
> > >
> > > First, we feel that our main contribution (hierarchical interpretations) is, in fact, novel. No paper, including the CD paper, has considered this, with much recent effort being put into (non-hierarchical) heat maps (we cite 13 recent papers in our related work that do this). Our algorithm for hierarchical importance is also independent of CD, and could be applied for any suitable patch/phrase importance score. Our experiments show that applying our hierarchical interpretation algorithm (our main contribution) to CD, yields higher user trust, and more insight into a model's predictive accuracy, than CD alone, or any of our other baselines.
> > >
> > > To echo our previous response, our point is that moving from heat-maps, such as Figure 5 in [1] or Figure 4 in [2] (two of our baselines), to hierarchies, such as Figure 2 in our paper, is not incremental. Rather, it is a simple algorithm that produces a novel form of interpretation (hierarchies) with validated empirical benefits.
> > >
> > > [1] https://arxiv.org/pdf/1612.08220.pdf
> > > [2] https://arxiv.org/pdf/1703.01365.pdf

---

### Official Review · AnonReviewer1 · 2018-11-02
**Contextual decomposition for general DNNs**

**Rating:** 6
**Confidence:** 4

**Review:**

**Summary**

In this paper, the authors extend an existing feature interpretation method for LSTMs to more generic DNNs. They introduce a hierarchical clustering of the input features and the contributions of each cluster to the final prediction.

**Strength**

1. Splitting information into binary groups at each layer is a neat approach to segregate interpretations.
2. Experiments are elaborate and cover the breadth of the proposed method well.
3. The paper is well presented and fairly easy to follow.


**Weakness**

1. Limited contributions in terms of novelty. This approach for RNNs is presented fairly well in the previous paper [Beyond Word Importance: Contextual Decomposition to Extract Interactions from LSTMs](https://arxiv.org/abs/1801.05453).
2. It seems that there is not enough justification for the modifications in beta and gamma made for convolution and pooling layers.

---

> ### Author Response · Authors · 2018-11-08
> **Thanks for your comments**
>
> Thanks for the helpful comments. We would like to address your concerns about the novelty of the work.
>
> “1. Limited contributions in terms of novelty. This approach for RNNs is presented fairly well in the previous paper.”
>
> As you correctly noted, one of the two contributions of this paper is to generalize CD from LSTMs to generic DNNs. However, we would like to clarify that the most important contribution of this paper is not generalizing CD, but introducing the concept (and implementation) of hierarchical importance for interpreting neural network predictions.
>
> The current state of the art for interpreting neural network predictions is word/pixel-level heat-maps, such as Figure 5 in [1]. Our main contribution is to introduce hierarchical interpretations, such as Figure 2 in our paper, and show that they improve over heat maps, the prior SOTA. In our human experiments, we compare our hierarchical interpretations (agglomerative CD, or ACD) against three non-hierarchical baselines, ultimately concluding that the hierarchy is in fact beneficial.
>
> We hope that we have clarified that hierarchical interpretations are the main contribution of our paper, and that this addresses your concern around the novelty of this work. To address this, we have (slightly) modified our introduction and method sections. We tried to make this clear throughout the paper and would welcome suggestions on how to avoid similar misunderstandings for future readers.
>
> “2. It seems that there is not enough justification for the modifications in beta and gamma made for convolution and pooling layers.”
>
> Thanks for pointing this out - we realize we omitted much of our justification in making the modifications for convolution and pooling layers. To address this, we have added some intuition in the fifth paragraph of section 3.1. Additionally, we have added a figure in page 27 of the supplement (Fig S5) showing the effect of our modifications to the update equations for convolution and ReLU layers.
>
>
> [1] https://arxiv.org/pdf/1612.08220.pdf

---

> ### Author Response · Authors · 2018-11-26
> **Response to review revisions**
>
> Thanks for your response. We see that you have responded to our comments by adding the sentence "They introduce a hierarchical clustering of the input features and the contributions of each cluster to the final prediction." onto the summary of our paper, and leaving the strengths, weaknesses and rating unchanged.
>
> In light of your updated summary, we feel your main concern should also be revisited. Your stated concern is that "This approach for RNNs is presented fairly well in the previous [CD] paper". Our main contribution, hierarchical interpretations, was not presented in the CD paper, or in any other paper, and is independent of CD (it can be applied to any phrase/patch importance score). The bulk of recent work in this area has focused on (non-hierarchical) heat maps (we cite 13 recent papers in our related work that do this). Our experiments show that applying our hierarchical interpretation algorithm to CD, yields higher user trust, and more insight into a model's predictive accuracy, than CD alone, or any of our other baselines.
>
> To be clear, our point is that moving from heat-maps, such as Table 1 in [1] (the CD paper),  Figure 5 in [2], or Figure 4 in [3] (our three baselines), to hierarchies, such as Figure 2 in our paper, is not incremental.
>
> We responded to concern #2 above, and also modified paragraph 5 of section 3.1 and added a figure on page 27 of the supplement (Fig S5). Did this address your concern?
>
> [1] https://arxiv.org/pdf/1801.05453.pdf
> [2] https://arxiv.org/pdf/1612.08220.pdf
> [3] https://arxiv.org/pdf/1703.01365.pdf

---

> > ### Comment · AnonReviewer1 · 2018-11-26
> > **Response to revision**
> >
> > Thanks for updating the paper and addressing my concerns, mainly regarding the novelty.
> > I will change my rating accordingly.

---

### Official Review · AnonReviewer2 · 2018-11-04
**Interesting hierarchical approach to explainability**

**Rating:** 7
**Confidence:** 3

**Review:**

This paper proposes a hierarchical extension of contextual decomposition. The approach is validated in qualitative examples and a small scale usability study

Quality,
The paper is well motivated. Contextual decomposition is briefly described but detailed enough to self-contained. The experimental evaluation produces usability evidence. Uncertainty could have been better explained,

Clarity,
The main methodological contribution (hierarchical CD) is well motivated but only  provided in the form of an algorithm. Could have been more precisely described and optimality discussed.

Originality & significance
The work builds heavily on CD but has the hierarchical extension is original and significant.
Uncertainty estimates could have improved the significance of the usability study

pros and cons
+ interesting problem
+ well-motivated algorithmic extension of CD
- uncertainty of usability experiment?

---

> ### Author Response · Authors · 2018-11-08
> **Thanks for the feedback**
>
> Thanks for the helpful comments and positive feedback. We’re glad you agree that the hierarchical notion of importance is important and well validated. We address some of your concerns below.
>
> “The main methodological contribution (hierarchical CD) is well motivated but only provided in the form of an algorithm. Could have been more precisely described and optimality discussed.”
>
> This is great feedback, we agree that we were missing a bigger picture, non-mathematical description of the algorithm. We have added a paragraph at the beginning of Section 3.2 giving intuition for our method before jumping into the technical details.
>
> “Uncertainty estimates could have improved the significance of the usability study.”
>
> We agree, and have updated the paper to include statistical significance results. We’ve provided the details below, but the summary is that most of the big jumps in our plots were significant, with the exception of ImageNet in plot A, where the results were only “suggestive” (p values ranged from 0.07 to 0.15). Overall, it seems like the benefits of ACD are in fact statistically meaningful.
>
> Statistical significance results summary:
>
> Identifying an accurate model (one-sided two-proportion t-test)
> Sentiment: gaps between ACD and IG, break-down are significant, ACD to CD is not
> MNIST: nothing is significant
> ImageNet: gaps between ACD and others are suggestive, but not significant (p values range from 0.07 to 0.15)
>
> Ranking trust in model (permutation test with mean rank test statistic)
> Sentiment/ImageNet: ACD’s mean rank is significantly higher than all other methods
> MNIST: ACD is significantly higher than break down, everything else is not.

---

### Public Comment · ~Shi_Feng1 · 2018-11-03
**Related (and likely concurrent) work**

Godin et al. just presented at EMNLP an extension of CD to CNNs: https://arxiv.org/abs/1808.09551
It should be noted that it was put on arxiv 2 months later than yours. But it would be interesting to compare the two, for example Godin et al. did not partition the bias.

---

> ### Author Response · Authors · 2018-11-08
> **Thanks for your interest in our work**
>
> Thanks for your interest in our work, Shi! I think it is worth clarifying that in both our work and Godin et al., the extension of CD is a relatively secondary contribution to the “meat” of the paper (for our work that is the hierarchical interpretation, for Godin et al. it is the analysis of character-level neural networks).
>
> Interestingly, our original extension of CD to CNNs was quite similar to Godin et. al. However, when we tried this on ImageNet we found that the results were both qualitatively bad, and often produced very large importance scores (indicating that something was “blowing up”). To fix this, there were two main changes we made (which is also where we differ from Godin et al.): (1) partitioning the bias in a conv/linear layer and (2) modifying the decomposition for the ReLu nonlinearity.
>
> We have added a paragraph to 3.1 and a supplementary figure (Fig S5) on page 27 both to show the difference between our approach and Godin et al. and to better motivate our generalized CD. Hopefully this can shed some light into the differences in behaviour in a vision setting.

---

### Author Response · Authors · 2018-11-08
**Modifications to paper in response to reviewers and commenters**

We thank the reviewers for their time and thoughtful comments. In response to their input, we have made the following changes to the manuscript:

1. To address  reviewers 2 and 3’s concerns about the motivation of our agglomeration procedure, we added a paragraph at the beginning of Section 3.2 to give more intuition on our agglomeration procedure, before delving into the precise, mathematical details.
2. To address reviewer 1’s questions about the motivation of generalizing CD to CNNs, we added the fifth in Section 3.1 to give more details.
3. To address Shi Feng’s questions about comparing with Godin et al.’s concurrent work, and reviewer 1’s comments about motivating our generalization of CD, we added Figure S5 on page 27 to illustrate rationale for equations in CD and compare against Godin et al.
4. To address reviewer 2’s concerns about uncertainty estimates for our usability study, we added statistical significance analysis for human experiments in the third and fifth paragraphs on page 8.
5. Minor modifications for clarity in response to the reviewers (e.g. in the introduction, method section)

We should note that the pdfdiff is showing more changes than we actually made. This is because we had to move one of our larger figures to keep things in place, which appears to have triggered a change, e.g. the bottom half of page 8. In these cases, the actual text and layout of the paper has not been altered.

---

### Meta-Review · Area_Chair1 · 2018-12-16
**Unanimous accept.**

**Confidence:** 4
**Recommendation:** Accept (Poster)

**Metareview:**

The paper receives a unanimous accept over reviewers, though some concerns on novelty exist. So it is suggested to be a probable accept.